# On the Infectious Causes of Neonatal Piglet Diarrhoea—A Review

**DOI:** 10.3390/vetsci9080422

**Published:** 2022-08-10

**Authors:** Magdalena Jacobson

**Affiliations:** Department of Clinical Sciences, Faculty of Veterinary Medicine and Animal Husbandry, Swedish University of Agricultural Sciences, 75007 Uppsala, Sweden; magdalena.jacobson@slu.se

**Keywords:** *E. coli*, *C. perfringens*, *C. difficile*, *Enterococcus* spp., TGE, PED, rotavirus, *Cystoisospora suis*, *Strongyloides ransomi*

## Abstract

**Simple Summary:**

This review is a summary of the current knowledge of the aetiology of neonatal porcine diarrhoea, including the early reports and the growing knowledge on the disease, some clinical features, a brief description of the suspected or known causes, pathogenesis and pathology, surveys made on the disease, and aspects of the methodology to be used in such microbiological surveys. Considerations of treatment and prophylaxis are beyond the scope of this paper.

**Abstract:**

The aim of this paper is to review current knowledge on the relationship between presumptive infectious agents and neonatal porcine diarrhoea (NPD). The literature provides information on the rationale for this causation, including the first mention, main understandings gained with respect to, e.g., pathogenesis, and the knowledge to date on the specific relationships. Further, surveys on the presence and relative importance of these pathogens in NPD are included and the methodology used to identify the causation are discussed.

## 1. Methodology

The literature search has been based on the key words (neonatal diarrhoea OR neonatal diarrhea) AND (pig* OR porcine OR hog* OR swine) OR (neonatal diarrhoea AND *Escherichia coli* OR *Clostridium perfringens* OR *Clostridium difficile* OR *Enterococcus* spp. OR TGE OR PED OR rotavirus OR *Cystoisospora suis* OR *Strongyloides ransomi*) using the data bases PubMed, Google Scholar and Web of Science. References given in this literature and in standard reference literature have also been considered.

## 2. Background

Neonatal porcine diarrhoea (NPD) was recognized as a serious problem in the late 1950s and 1960s with the emergence of the modern pig industry [1]. Over the years, various aetiological agents have been described. By the 1930s, intestinal disease in neonatal piglets kept on large ranches was associated with *Bacillus (Escherichia*, *E.) coli* [2] and later, the development of serotyping methods enabled the distinction of certain serotypes being related to neonatal diarrhoea [3].

In 1934, coccidiosis caused by the new species *Isospora (Cystoisospora*, *C.) suis* was identified as the cause of diarrhoea in pigs [4]. In 1946, outbreaks of a transmissible gastroenteritis (TGE), presumably caused by a virus that affected pigs of all ages and with high mortalities in neonates was described [5]. In the 1950s, a necrotizing enteritis in newborn piglets was associated with infection by *Clostridium (C.) welchii (perfringens)* type C [6], and in 1966, porcine strongyloidosis was reportedly an important cause of neonatal diarrhoea, with mortality rates approaching 75% [7].

In the 1970s, the syndrome “baby pig scour” emerged and a causal relationship with enterotoxigenic strains of *C. perfringens* type A was proposed [8]. Further, rotavirus was isolated and shown to cause diarrhoea in 2- to 28-day-old gnotobiotic pigs [9]. In 1972, an outbreak of epidemic diarrhoea in pigs >10 weeks of age was reported [10]. Subsequently, a disease resembling TGE in suckling pigs, but with lower mortality rates, was described. TGE virus was not detected, and the tentative name “epidemic diarrhoea type II” was proposed [11].

Going forward, neonatal diarrhoea not related to enterotoxigenic *E. coli* (ETEC), *C. perfringens* type C, TGE, or coccidiosis, was reported. Several potentially pathogenic, causative agents have been suggested, including *C. perfringens* type A [8,12,13], *Clostridioides (C.) difficile* [14], previously largely overlooked *E. coli* strains such as enteropathogenic *E. coli* (EPEC) [15], rotavirus [16], and members of the *Enterococcus (E.) faecium* species group (*E. durans*, *E. hirae*, *E. villorum*) [17,18,19,20].

NPD has been given various names: neonatal diarrhoea [1,21], neonatal colibacillosis [22], baby pig scours [8], enterotoxemia in baby pigs, infectious gastroenteritis of suckling pigs, neonatal haemorrhagic and necrotic enteritis, necrotizing enteritis, neonatal necrotic enterotyphlocolitis [23,24], epidemic diarrhoea type II [11], transmissible gastroenteritis [5], or new neonatal porcine diarrhoea (NNPD) [25].

## 3. Clinical Signs

The neonatal phase arbitrarily refers to the first few days, up to one week, after birth [26,27]. “Diarrhoea” may be defined as malabsorption of water or electrolytes [28], the passing of loose or watery faeces [29], shedding of aqueous or bloody faeces, and/or more frequent defecation [30], and a faecal water content exceeding 80% [31] or a dry matter of ≤18% [32]. The initially semi-solid faeces in healthy, newborn piglets must be distinguished from diarrhoea [33]. The clinical signs vary with the pathogenesis, the infectious load, and the innate and passive immunity of the piglet. It is rarely possible to relate the signs to a specific aetiology, but some features may be indicative [34].

Piglets affected by ETEC may develop severe, secretory diarrhoea within 1–2 h after birth [1], displaying profuse, non-haemorrhagic, watery diarrhoea, nausea, abdominal cramps, and shivering. The piglets rapidly become dehydrated, emaciated, and weak. The morbidity and mortality may range from a few piglets in a few litters, to all piglets in almost all litters [1,33,35]. Pigs may recover quickly, since there is little physical damage to the intestine [22]. The possibly malabsorptive diarrhoea caused by EPEC affect pigs 2–4 days p.i. and is noted as voluminous, mucoid, and less viscous faeces. The recovery period will be prolonged due to the need for regeneration of sufficient absorptive epithelium [22]. The pigs initially remain active and alert but 2 weeks later, seem stunted [36], and are frequently described as “poor doers” [37].

*Clostridum* spp. commonly affect 1 to 3-day-old pigs [13,14,38]. The necrotic lesions caused by *C. perfringens* type C result in haemorrhagic diarrhoea and an increased mucosal permeability. Pigs being naïve to the infection rapidly become moribund [22,33], whereas chronically affected animals suffer from prolonged, intermittent diarrhoea, and eventually may die [38,39]. Reportedly, *C. perfringens* type A (CpA) causes scouring of fluid, foamy to creamy yellow diarrhoea with flecks of blood, and high morbidity but negligible morality rates. The pigs remain vigorous and recover spontaneously [13,38,40]. *C. difficile* affects the large intestine and may cause increased mortality rates, dyspnea, emaciation, increased abdominal distention, scrotal oedema, and semifluid, yellowish-brown faeces [14,38].

Among the enterococci species, *E. durans* causes diarrhoea with high morbidity, but low mortality in 2 to 14-day-old pigs. Some individuals become severely stunted, but others continue to suckle and gain weight [17]. Similarly, *E. hirae* may induce diarrhoea, weight loss, and rough hair coat in 1–5 day old piglets [18,20,41]. In one study, more than half of the piglets in 1 to 3-day-old litters were affected, and the mortality was <10% [42].

In naïve herds, TGE affect pigs of all ages within 24–72 h post exposure with profuse, watery diarrhoea, vomiting, dehydration, and high mortality in neonatal pigs [5,43]. In endemically infected herds, long-term, mild diarrhoea with low mortality rates in 1 to 4-day-old pigs, has been seen [43,44,45]. Porcine epidemic diarrhoea, PED, was initially described as a “mild form of TGE” [46] but later, outbreaks clinically undistinguishable from TGE have emerged [47,48,49]. Experimentally, rotavirus rapidly induced signs similar to TGE in gnotobiotic pigs. In a clinical setting, mild, watery to creamy, whitish-yellowish diarrhoea usually occurs in 1 to 4-week-old pigs, with high morbidity but low mortality, being referred to as white scour, three-week scour, or milk scour [34,50,51,52,53,54]. However, rotavirus group C has also been demonstrated in <3 days old piglets, as well as in fatteners [53].

Similarly, *Cystoisospora suis* is usually shed in 5–10 day old piglets [55], but experimentally, anorexia, profuse, yellow-to-gray, foetid malabsorptive diarrhoea, the shedding of large amounts of mucus and sloughing of epithelial cells, followed by constipation, may be noted 3–4 days p.i. [4,21,22,55]. *Strongyloides (S.) ransomi* may cause necrotizing lesions and increased permeability of the intestinal mucosa [22]. Experimentally, profuse diarrhoea, anorexia, anaemia, severe itch, muscle and abdominal pain, growth retardation, and mortality rates approaching 50% are seen 3 days p.i. up to two weeks of age [56].

## 4. *Escherichia coli*

*E. coli* is part of the microbial intestinal flora and may broadly be classified into commensal microbiota *E. coli*, enterovirulent *E. coli*, and extraintestinal *E. coli* [57]. The strains may be identified by seroagglutination to a panel of diagnostic sera displaying reactivity to known O (lipopolysaccharide), H (flagellar), K (polysaccharide), and F (fimbriae) antigens [58]. *E. coli* may also be grouped according to their pathotype, showing distinct clinical patterns, epidemiology, pathogenesis, and O:H serotypes. Several groups have been implicated in diarrhoeal disease: ETEC and EPEC that are major causes of diarrhoea in neonates [37,59], enteroinvasive *E. coli* (EIEC), enterohemorrhagic *E. coli* (EHEC), and enteroadherent or enteroaggregative *E. coli* (EAEC/EAggEC) [57,60]. In addition, diffusely adhering *Escherichia coli* (DAEC) [57,61], adherent-invasive *E. coli* (AIEC) [57], and necrotoxigenic *E. coli* (NTEC), may be distinguished [37]. Three pathotypes are of particular importance in the pig: ETEC, EPEC, and EHEC, the latter also including VTEC (verotoxin-producing *E. coli*) or STEC (Shiga toxin-producing *E. coli*), [37], although some authors distinguish VTEC as a separate pathotype [62]. The terms VTEC and STEC can be used synonymously.

ETEC colonises the small intestine and is a well-known cause of neonatal colibacillosis. It adheres to the intestinal mucosa by specific adhesins (fimbriae) and causes diarrhoea by the action of enterotoxins that induce hypersecretion of water and electrolytes [35,37]. ETEC carrying the F4, F5, F6, F18, and F41 fimbriae with the capability to produce the LT (heat-labile) and STa and STb (heat-stable) toxins is considered as the most important aetiology of NPD worldwide, although strains carrying the adhesin AIDA-I may be of increased importance [12].

EPEC may colonise the entire intestine [36,60]. The term initially distinguished serotypes that caused enteric, rather than extraintestinal, infections [59,63]. Later, serotyping was found to have a poor specificity and “true” or “typical” EPEC (tEPEC) may be defined by their genetic “makeup” [64], failure to produce enterotoxins, and lack of invasive properties [36,37]. Ultrastructural lesions are characterized by the destruction of microvilli (effacing) and intimate attachment to enterocytes mediated by the adhesion intimin, referred to as attaching and effacing *E. coli* (AEEC) [36,37]. EPEC can be subdivided into class I (tEPEC), exhibiting localized adherence, and II (atypical or aEPEC), with diffuse or no adherence. DAEC strains expressing the adhesin AIDA-I belong to aEPEC [37,57]. EPEC exerts its pathogenicity by the loss of the absorptive surface and cytoplasmic degeneration, causing malabsorptive diarrhoea [36] and secretion of chloride [37].

EHEC [65] acts by the use of potent Shiga-like or verotoxins (Stx/Vtx) and may be subdivided into: 1. EHEC, associated with haemorrhagic colitis and haemolytic uraemic syndrome in humans, 2. EHEC or EHEC-like isolates with similar properties but associated with animals, and 3. other Stx-producing strains (aEHEC). In the latter group, strains producing the Stx2e toxin that are responsible for oedema disease in weaned pigs are included. These strains may also carry features associated with ETEC, such as F4 fimbriae and the ability to produce LT and ST toxins [66]. EHEC attach to epithelial cells by fimbriae and, similar to EPEC, exhibit localised adherence, inducing attachment and effacing lesions [37,66]. Experimentally, EHEC infection involves the caecum and colon in pigs [60]. The strain O141:K85, frequently isolated with oedema disease, may also be associated with neonatal diarrhoea [66].

DAEC exhibits diffuse adherence patterns and was previously subdivided into DAEC expressing the adhesin Afa/Dr, and strains not expressing Afa/Dr but expressing AIDA-I (aEPEC). The intestinal DAEC is involved in diarrhoea in children, but the intestinal sites colonized by Afa/Dr DAEC remain to be determined [57].

NTEC produces the toxin Cytotoxic Necrotizing Factor (CNF) 1 and 2. CNF1 has been demonstrated in pigs of various ages, including <15-day old piglets with diarrhoea. CNF 2 has not been demonstrated in pigs [67]. Some isolates contain adhesins with the ability to adhere to intestinal villi (Afa, P fimbriae type III, S or F1C fimbriae) [68,69]. NTEC has also been implicated in extraintestinal infections [37].

EIEC has a predilection for the colonic mucosa and invades the epithelial cells in humans, causing a dysenteric form of illness with fever, abdominal cramps, malaise, toxaemia, and watery diarrhoea with scanty flecks of blood and mucus [60,70].

Pathogenic strains of EAEC/EAggEC colonize the small intestine and/or colon [61] and may cause acute, watery, persistent diarrhoea in young children, and food-borne outbreaks in humans. EAEC does not show ST, LT and does not invade epithelial cells [57,60]. It adheres to the mucosa by the aggregative adherence fimbria (AAF/1, II och III, and type IV pili) and forms typical “stacked-brick” microcolonies, described as an aggregative adherence pattern. The lesions induced include microvilli vesiculation, enlarged crypt openings, and increased epithelial cell extrusion. However, not all strains carry the genes encoding for these factors. The EAEC carries a great diversity of adhesins and produces a variety of enterotoxins and cytotoxins [57,61,71], including EAST-1 that is also found in the EHEC/STEC, ETEC, DAEC, and EPEC pathotypes [71]. EAST-1 acts by stimulating guanylate cyclase, similar to STa [61]. EAEC-typical strains carry the gene *aggR*, but atypical strains may also cause diarrhoea [61].

## 5. *Clostridium* spp.

*C. perfringens* constitutes part of the intestinal flora in healthy animals. Based on the production of major toxins, it may be divided into five toxinotypes, namely A to E. All types produce the α-toxin, as well as the enterotoxin that causes food poisoning in humans [38].

*C. perfringens* type C (CpC) is a well-known pathogen, seemingly confined to certain geographical areas [22,39]. The disease affects the jejunum and ileum, with haemorrhagic necrosis of the villi, sometimes involving all layers of the intestinal wall. Occasionally; lesions are also noted in the caecum and proximal colon [8,24,38]. In naïve piglets; mortality may reach 100%. In endemically infected herds, the sows will provide protective immunity through colostrum and clinical signs will merely be noted in gilt litters [38]. Besides α-toxin, all strains also elaborate β-toxin, which is the most lethal and necrotizing component [38]. Both the bacteria and the toxin seem to be necessary in the pathogenesis [39]. Several predisposing factors have been suggested, including high gastric pH and low tryptic activity in newborn piglets, trypsin inhibitors in the colostrum, and decreased intestinal motility [24,33,38,39].

Most strains of *C. perfringens* type A (CpA) produce the toxin CPB2 that may be involved in NPD [72] although its role is controversial [38,73]. Moreover, the enterotoxin has been suggested to cause disease in a 1-week-old piglets [74]. One experimental study describes congestion, focal haemorrhage, necrosis, villus atrophy, loss of epithelial cells in the small intestine, and serositis 2–4 days p.i. [40]. Attachment and invasion is uncommon, and epithelial necrosis is rarely described [38]. Currently, there is no way to differentiate between disease-causing strains and strains that are part of the commensal flora [40,73]. Some unidentified virulence factors may be present; however, this is speculative, and until more is known about possible associations to NPD, no definite conclusions can be drawn [23].

*C. difficile* is ubiquitous in the intestine, but rarely causes disease [75]. High numbers of the bacterium may be isolated in sick piglets, but the role in NPD is unclear [76]. Disease was first described during experimental infection of 11 to 31-day-old gnotobiotic pigs, which subsequently developed reduced appetite and mucoid diarrhoea with specks of blood [77]. In 1998, clinical cases were described in newborn piglets from a newly established, high-health herd suffering from typhlocolitis and inconsistently, diarrhoea, with high mortality rates in piglets [14]. The presence of toxins, mesocolonic oedema and volcano lesions in the colon are regarded as hallmarks of the disease [78,79]. The bacterium produces the enterotoxin TcdA and the cytotoxin TcdB, and usually, a large number of affected piglets are toxin-positive [38,80]. However, the toxins are also detected in healthy piglets [73].

## 6. *Enterococcus* spp.

The *Enterococcus* species is commensal in the intestinal tract, but members of the genus group III [81] have been implicated as possible causes of diarrhoea in suckling animals [19,81,82]. The pathogenesis remains unknown, but may be related to a decreased activity of the brush border enzymes and interference with digestion and absorption. Misidentification among the species may occur [19,83], but three species of particular importance have been associated with neonatal diarrhoea in piglets: *E. villorum* [19], *E. durans*, and *E. hirae.* By histology, the predominant lesions caused by *E. durans* were the result of the extensive colonisation of the small intestinal enterocytes by Gram-positive cocci and a few sloughed enterocytes [17]. Experimentally, diarrhoea related to colonisation of the mucosa, necrosis of the villous epithelium, and villous atrophy has been induced [84]. Similarly, lesions related to *E. hirae* [18,20,41,83] consisted of flaccid and dilated intestines with liquid to creamy content [18,20,85], and by histology, numerous enteroadherent Gram-positive cocci were associated with villous epithelial damage and atrophy [18,41,83,85]. In studies on the intestinal microbiome in piglets with diarrhoea, the amount of *Enterococcus* was up to 24 times more abundant in the intestinal content from sick pigs as compared to healthy animals, and it was concluded that *Enterococcus* sp. has an important role in pathogenesis [82,86].

## 7. Viral Infections

Coronaviruses are classified as Alpha-, Beta-, Gamma-, and Delta-coronavirus. TGEV and PEDV are members of the genus Alphacoronavirus, while Gammacoronavirus has never been detected in pigs [48]. However, new mutation and recombinant variants occur [87], and an outbreak of diarrhoea in sows in the US with 30–40% mortality in piglets was caused by Deltacoronavirus [48], whereas a new variant of Alphacoronavirus was reported from China [88], later designated SADS-CoV [89]. Transmission is considered to occur through the faecal-oral route by direct co-mingling of pigs or by fomites [46,90], but airborne transmission [90] and transmission by other animals, e.g., dogs, has also been suggested [91]. The persistence at herd level is related to management and herd factors [43,92].

TGE mainly affects the jejunal and ileal villus epithelial cells. Macroscopically, thin-walled, translucent intestines are observed, caused by villi atrophy and crypt hyperplasia, resulting in malabsorption. The severity of lesions and clinical signs is age-dependent [44,93]. The virus is easily transmitted. In endemically infected herds, most pigs will be immune, and diarrhoea with typical lesions may only be present in a subset of pigs [34,43,45]. Since the 1990s, TGE has declined, possibly due to cross-protection induced by the immunologically similar porcine respiratory corona virus [43,46,94].

The first outbreaks of PED described were shown to be caused by an antigenically distinct virus [10,11,46]. In a Belgian outbreak of epizootic diarrhoea in pigs of all ages, high mortality rates were noted in piglets <1 week old. Experimentally, watery diarrhoea developed within 36 h [95]. Thereafter, the occurrence diminished [46]. In 2013, a highly virulent PEDV emerged in the US, affecting pigs of all ages, with up to 95% mortality in suckling pigs. The clinical signs and intestinal lesions were similar to those caused by TGE [46,90], although it infrequently also involved the colon [92]. In 2014, a new variant, initially only affecting sows, was described [49,90,92]. The US strains were highly similar to each other and to a Chinese strain [47,90]. One year later, an outbreak of PED with watery diarrhoea in all age groups was reported in Germany. The strain had a high similarity (99.5%) to the US strains, was less similar (97.1%) to the strains previously isolated in Europe [96]. In recent years, PED has progressively disappeared in Europe [92].

Rotavirus (RV) is classified into groups A–J (I and J being provisional species) based on antigenic properties. Of these, RV groups A, B, and C are most commonly identified and may also have zoonotic potential [97]. RVA, B, C, E, and H have been described in pigs, and A, B, and C may be associated with NPD [51,53,97,98,99]. Experimentally, 2 to 28-day-old gnotobiotic pigs developed depression, anorexia, vomiting, profuse diarrhoea, and a rapid deterioration of overall condition [9,51]. In naïve herds, the clinical signs commenced 1–2 days after birth, with a reported within-litter morbidity of 100%, and case-fatality rates of 5–10% [99]. The age of onset seems related to the level of lactogenic immunity, providing strong passive protection that may suppress the development of an active immunity [54,97]. Subclinical infections are common [9,100]. The virus infects the epithelial cells at the tips of the small intestinal villi, causing desquamation, villous atrophy and fusion, malabsorption, and thin-walled intestines, sometimes resembling lesions caused by TGE [9,54,101]. Further, a secretory component has been described [97]. Co-infections of RVA, RVB, and RVC are also noted [53]. The large genetic diversity might explain the varying clinical picture, as well as the hitherto variable effect of vaccination [97].

In addition, the development of improved diagnostic methods have resulted in the identification of a number of other viruses suggested to be involved in disease; however, thus far, their involvement in NPD is not well defined [102,103,104,105,106,107].

## 8. Parasitic Infections

Due to the improved housing and hygiene employed in modern pig production, the prevalence of many parasitic infections has been altered. Two parasites are reportedly related to NPD, namely *C. suis* and *S. ransomi* [108,109].

*C. suis* is frequently found in modern swine production [108,110]. To differentiate *C. suis* from *Eimeria* spp., which is commonly identified in faeces from adults, the in vitro sporulation of the oocysts is necessary [108]. In the pen, sporulation requires at least four days [4], but experimentally, *C. suis* sporulated within 16 h at 30 °C [111]. Following ingestion, the parasite penetrates the small intestinal enterocytes and passes through four developmental stages before being released and shed as unsporulated oocysts [112]. By histology, prominent villous atrophy, particularly in the jejunum and ileum, mild erosions and necrosis of the lamina propria, with neutrophilic infiltrates and intracellular parasites, are seen four to 10 days p.i. [21,34]. The lesions may extend into the colon and resolve within 14 days [21]. The disease is dose-dependent, and inoculation with 200,000 oocysts resulted in emaciation, lethargy, profuse diarrhoea, and necrotic enteritis, whereas 400,000 oocysts caused the death in 10 out of 12 pigs within 3–4 days [21].

Experimentally, *S. ransomi* may cause profuse diarrhoea in 3-day-old piglets [56]. The prepatent period is stated to 3 to 4 days by the lactogenic route of transmission, and otherwise, up to 9 days. Thus, piglets less than 4 days of age will rarely display diarrhoea [7,113]. The life cycle includes a free-living development, followed by percutaneous or oral infection and migration via the lymphatic or haemato-tracheal routes, and development of the adult female in the proximal small intestine. Further, inhibited somatic larvae may reside in the mammary adipose tissue, and following reactivation, larvae are transmitted by colostrum to the piglets. Eggs of *S. ransomi* are difficult to differentiate from those of free-living nematodes. The occurrence is usually sporadic, with low prevalence, and the parasite is not considered as a major pathogen, although single, clinically significant outbreaks may still occur [22,108,109,113].

## 9. Surveys

The results from different surveys vary with the aim of the study, the number of samples, the analyses included, the diagnostic techniques used, and with the current knowledge at the time of the investigation. The studies are listed in Table 1.

In the early 1980s, TGE was a major finding, with a reported prevalence of 50% of the outbreaks and 59% of the pigs [34,43]. ETEC and *C. suis* were commonly detected, and with lower prevalence, cryptosporidia, PED, rotavirus, *Campylobacter coli*, adenovirus, CpC, CpA, and *Salmonella* spp. were demonstrated [12,34,43]. The age of the pigs ranged from <5 days in ETEC infections, 5–15 days in *C. suis* infections, and >10 days in rotaviral infections [34]. In 1989, an outbreak of NPD was associated with the presence of CpA enterotoxin and the absence of rotavirus and TGE. By cultivation, mixed *E. coli*, enterococci, and *Bacillus* sp. were also demonstrated [13].

In the 1990s, endemically occurring TGE and PED were still major findings in single outbreaks [45,46]. Additionally, nonhemolytic *E. coli*, Gram-positive cocci, *C. perfringens*, *C. difficile* [14], *C. suis*, and ETEC have been reported in single herds [45]. Further, a Spanish survey on the various *E. coli* pathotypes identified ETEC in 15% of the diarrhoeic piglets and NTEC in three piglets [67].

During 2000–2010, studies in the US and Japan on 1 to 7-day-old, diarrhoeic piglets found rotavirus (30%) and *C. difficile* (30%), respective rotavirus (82%), and ETEC (13%) to be the most common, whereas *E. durans*, TGE, PED, haemolytic *E. coli*, CpC, sapovirus and coccidia were detected to a lesser degree [114,115]. Contrariwise, a German study did not detect any rotavirus in this age category, the most common pathogens instead being ETEC (10%) and *C. suis* (7%), whereas coronavirus, *C. parvum*, and *S. ransomi* were not found [30].

In the last decade, CpA, *C. difficile* and its toxins, and/or rotavirus, are commonly found in neonatal pigs, irrespective of health status [27,76,116,117,118,119]. In one study, RVA was the only pathogen significantly associated with diarrhoea [119]. However, conflicting results are common: nine piglets were all positive for RVB, whereas TGE, ETEC, *C. difficile*, and *C. perfringens* were not found [52], or ETEC was identified as the sole pathogen in most cases, whereas less than 10% were positive for CpA, rotavirus, *C. difficile*, *C. suis*, TGE, *Salmonella* spp., *C. parvum*, and *E. durans* [73]. Cases submitted in 2010 were more likely to be diagnosed with CpA, as compared to those submitted in 2001 to 2007 [73]. Pathogens reported to be less frequent, or not detected, include coccidia, haemolytic *E. coli*, ETEC, NTEC, EPEC, EDEC, PED, CpC, *Cryptosporidium* sp., *Salmonella* spp, and TGE [27,76,116,117,118,119].

In Denmark, 3 to 7-day-old pigs from four herds, including 50 healthy and 51 piglets that had suffered from diarrhoea for at least one day, were euthanized and submitted for necropsy. CpA was frequently found, being twice as common in the healthy controls. Haemolytic *E. coli* was found in three, and ETEC in one, diarrhoeic piglet. CpC was found in four pigs in one herd and *C. difficile* in 4% of the healthy piglets [25]. By PCR and NGS, RVA was found in 13% of the diarrhoeic pigs, and RVC was found in one pooled faecal sample [120]. Associated to villous atrophy and epithelial lesions, enteradherent non-ETEC were noted in 33% of the diarrhoeic and 14% of the healthy piglets, and enteroadherent *Enterococcus* spp. was found in 27% of the diarrhoeic and 2% of the healthy pigs [85]. The genus *Enterococcus* was 24 times more abundant in diarrhoeic piglets. Piglets born to gilts had 25 times higher odds of having NNPD [86]. 

**Table 1 vetsci-09-00422-t001:** An overview of various surveys on the causes of neonatal porcine diarrhoea (NPD), including the year of the study, age of the animals, the number of animals/herds included, the specimen used in the study, the presumptive pathogens found in piglets with NPD and (if applicable) in healthy controls, pathogens investigated for but not found, and the references to each study.

Country, Year	Age	No of Animals/Herds	Specimen	Findings, Diarrhoeic Piglets	Findings, Healthy Controls	Investigated but Not Demonstrated	Reference
Canada 1977–1981	1–15 d.	749/325	Carcasses	TGE 52%; ETEC 22.4%; *C. suis* 15.3%; RV 9.2%; CpC 0.4%; AV 0.3%	N.A.	N.A.	[3]
England 1981–1985	N.A.	N.A./85 240/78	Anamnestic information Carcasses	*E. coli*; RV; *Salmonella* spp.; TGE 25%; haemol. *E. coli* 5.8–32%	N.A.	N.A.	[43]
UK 1982	3 d.–3 w.	116/3	Carcasses	CpA 10.3%; coccidia, cryptosporidia; PED; RV; *C. coli*; ETEC rarely	N.A.	N.A.	[12]
USA 1989	1-2 d.	3/1 10/1	Carcasses Faeces	CpA; ETEC; enterococci; *Bacillus* spp.	CpA	TGE; RV N.A.	[13]
Spain 1986–1991	<15 d.	149/65	Faeces	ETEC 15%; VTEC 2.2%; NTEC 2.2%	0%	N.A.	[67]
Canada 1995	3 d.	14/1	Carcasses Faeces	TGE 33%; ETEC 15%; coccidia 13%	N.A.	N.A.	[45]
USA 2000	1–7 d.	100/33	Carcasses	RV 36%; *C. difficile* 29%; *E. durans* 5%; TGE 3%; haemol. *E. coli* 3%; CpC 2%	N.A.	N.A.	[114]
Germany 2001	1–7 d.	33–41/24	Faeces	ETEC 9.8%; *C. suis* 7.3%	N.A.	Coronavirus; RV; *S. ransomi*	[30]
Japan 2001–2003	0–7 d.	60/14	Faeces	RV 81.7%; ETEC 13.3%; Sapovirus 3.3%;	N.A.	Coccidia; *C. parvum;* TGE; PED; CpC; *Salmonella* spp.	[115]
Denmark 2011–2014	3–7 d.	101/4	Carcasses	Haemol. *E. coli* 6%; non-haemol. *E. coli* 47%; *E. coli* serotype O8 4%; EHEC 8%; CpA 35%; CpC 6%; RVA 13%; *Enterococcus* spp. 45%	Non-haemol. *E. coli* 48%; *E. coli* serotype O8 6%; CpA 70%; CpC 2%; *C. difficile* 4%; RVA 2%; *Enterococcus* spp. 8%	Coronavirus; *C. suis*; *Cryptosporidium* spp.; *Giardia* spp., *S. ransomi*	[16,25,85,86,120]
Sweden 2011–2015	1–6 d.	69/10	Carcasses	Enteroadherent *Enterococcus hirae* 36%; haemol. *E. coli* 8%; ETEC 4%; CpA 76%; *C. difficile* 100%; RVA < 20%, RVC < 10%	CpA 68%; *C. difficile* 100%; RVA 10.5%	CpC, Coronavirus; *C. suis*; *Cryptosporidium* spp.; *Giardia* spp., *S. ransomi*	[18,20,107]
Germany 2017	1–7 d.	555/205	Faeces	CpA 59%; *C. difficile* 56.1%; pathogenic *E. coli* 38.6%; RVA 35%; PED 2%	N.A.	TGE	[76]
Spain 2017–2018	1–7 d.	215/31	Faeces	RVA 51.6%; RVB 9.3%; RVC 39.1%; TGE 2.8%; PED 2.3%; CpA 70.7%; CpC 3.3%; pathogenic *E. coli* 44%; *C. difficile* ~34%	RVA 31.8%; RVB 4.9%; RVC 36.4%; PED + TGE 2.3%; CpA 79.5%; CpC 1.1%; *C. difficile* ~29.5%		[119]
Spain 2018	1–7 d.	327/109	Faeces	CpA 89.9%; RVA 43.1%; PED 3.7%; ETEC 8.3%; EPEC 1.8%	N.A.	CpC, TGE	[27]

TGE = transmissible gastroenteritis; RV = rotavirus; RVA = rotavirus type A; AV = adenovirus; ETEC = enterotoxigenic *Escherichia coli*; CpC = *Clostridium perfringens* type C; CpA = *Clostridium perfringens* type A; *C. suis* = *Cystoisospora suis*; Haemol. *E. coli =* haemolytic *Escherichia coli*; *C. coli = Campylobacter coli*; *C. difficile* = *Clostridium difficile*; *E. durans* = *Enterococcus durans*; *C. parvum* = *Cryptosporidium parvum;* N.A. = not available.

Concurrently, 50 diarrhoeic and 19 healthy piglets from 10 Swedish farms were euthanized immediately before necropsy. The diarrhoea had commenced within one day before submission, and the majority of the piglets were ≤one day old. Lesions related to *E. hirae* were found in 60% of the diarrhoeic pigs from six herds and in none of the healthy pigs, with a higher proportion of *E. hirae* in the diarrhoeic pigs (*p* = 0.014). ETEC was only found in two piglets with diarrhoea, and Stb:EAST1:AIDA positive *E. coli* was found in one diarrhoeic and one healthy pig. CpA carrying the β2-toxin gene was found in all but one piglet, and following spore selection, *C. difficile* was isolated from all piglets. By metagenomics, rotavirus type A and C were found in pooled samples from both healthy and diarrhoeic animals in one herd, and RVA was found in diarrhoeic samples from one other herd (10%). CpC, EPEC, EAggEC, TGE, PED, *C. suis*, nor any other parasites, were demonstrated [18,20,107].

## 10. Discussion

The lack of an aetiological diagnosis in some animals is commonly reported [30,73,116,117], possibly due to the submission of suboptimal samples [34,73]. Many studies are based on faecal samples submitted for routine diagnostic purposes [30,104], and the investigation is dependent on the often scarce anamnestic information provided [30,34], e.g., “diarrhoea before weaning” [67,104]. Numerous biases may thus be present. Pigs in the acute stage of disease have the highest pathogen load and are therefore, the most likely to yield useful information [51,73]. Thus, untreated piglets having displayed typical clinical signs for less than one day should be selected for examination [20,22,34,116].

In many studies, several presumptive pathogens are identified [34,54,76], and some authors relate the mere presence of a specific pathogen to the disease [30,76,115,116]. However, the sole detection of an agent does not necessarily indicate a causal relationship [34,116]. Further, it is nearly impossible to state a diagnosis based on clinical signs [34,43,45,76], and with few exceptions, the colour of the stool is more likely reflect the feed ingested [9,22]. Adhesion is believed to be an initial and pivotal event in the pathogenesis, enabling the microbe to survive and persist despite peristalsis and the host defence mechanisms [57,85]. A definite diagnosis will thus be achieved following the histological examination of adequate specimens and the demonstration of heavy colonization by bacteria in close proximity, or adhered to, the mucosa [70,121], as shown, e.g., by studies on diarrhoea associated to *E. coli* [34,87], CpC [24,33,38,39], *E. hirae* [18,85], *E. durans* [17,84], and *E. villorum* [19]. 

The severity of disease varies with the extent of the lesions, and minor lesions will be compensated for by the normal function of other intestinal parts. Thus, the presence of diarrhoea indicates that intestinal alterations of sufficient severity are in place and may be detected [22]. However, a prompt post-mortem examination is necessary, since the macroscopic appearance of the digestive tract is lost within 6 h of death, and the desquamation of villi are already noted already within 1½ h [20,22,33]. For example, the acute intestinal lesions caused by rotavirus can be lost due to autolysis within 15 min of euthanasia [53].

Pathogens present in low numbers in the intestine of healthy animals may also be superimposed on lesions primarily caused by other agents [21,38,54,115,117]. Further, the extent of bacterial proliferation after death is substantial [33] and may bias the investigation. However, although several pathogens could be detected at the herd level, infectious diarrhoea in a single, newborn piglet is usually related to a single pathogen [27,34,115].

The diagnostic methods used vary considerably and often preclude comparisons and overall conclusions to be drawn. For example, not all ETEC strains are haemolytic, and diagnosis based solely on this feature may introduce false negative results [3,114,116]. Similarly, in studies on *Clostridium* spp., the inclusion of spore selection in the diagnostics will yield substantially different results [13,20]. Further, some studies address a single pathogen only [30], which may be relevant in prevalence studies, but less so in studies on a disease syndrome.

The relative importance of various pathogens seems to have changed over time. TGE, *C. suis*, ETEC, and CpC are well-known, major causes of NPD, but their clinical importance seems to have decreased [27,34,73,76,115]. The infrequent isolation may be due to, e.g., the implementation of vaccination programmes, natural immunity, or improved hygiene routines [20,22,27,30,37,42,73,115,116,119]. Instead, other pathogens such as *E. coli* carrying the adhesin AIDA-I, EPEC, and *Enterococcus* sp. are suggested to be of increased importance [18,37].

The common understanding of *C. perfringens* type A being a significant pathogen is difficult to substantiate, as the agent belongs to the normal intestinal flora [27,73,116]. The mechanisms involved in the infection are unclear, and no evidence for adhesion has been presented, nor does its presence correlate with any microscopic lesions [85]. Specific diagnostic criteria is lacking, and the current diagnosis is mainly made relying on the exclusion of other enteric pathogens, along with the detection of large numbers of *cpb2*-positive *C. perfringens* and its toxins in the small intestine. However, this is also a common finding in healthy neonatal pigs [20,38,73,116]. The lack of association suggests that CPB2 might not be a virulence factor, and that CpA is not associated with NPD [29,73,117]. It is thus possible that in the absence of other presumptive pathogens investigated, NPD is erroneously attributed to CpA [23,117].

Similarly, *C. difficile* is part of the normal intestinal flora in piglets [78,79] with an equal prevalence in both healthy and diarrhoeic pigs [117,119], and the toxins are also found in healthy pigs [116]. The disease supposedly occurs following overgrowth of the bacteria in the absence of an established microflora in, e.g., piglets less than 5 days of age, or in pigs treated with broad-spectrum antibiotics. A probable diagnosis requires demonstration of the toxins TcdA and/or TcdB and the presence of typical histopathologic lesions in the spiral colon [78,79].

The role of rotavirus as the primary pathogen in NPD is debated [9,97,99,100,119,122], since the diagnosis is often based on the mere presence of the virus [27], which, however, is also common in healthy animals. Likely, the immune status of the sow influences the age of onset and severity of disease [50]. Usually, mild diarrhoea of a few days duration is seen in suckling piglets >10 days of age [22,30,34,50,54,110], although severe disease may occur in animals exposed to high viral load, dual infections, poor environmental conditions, or poor lactogenic immunity [54,99,100,115,118]. Further, the cross-protection between the various groups of rotavirus and RVA genotypes seem to be limited [52,115]. The diagnostic methods used also varies considerably and most probably influences the sensitivity [30,73,118,120]. In recent studies based on necropsies and whole-genome sequencing, rotavirus was only sporadically found in piglets <7 days of age [107,120].

TGE is seemingly of less importance today. However, in endemically infected herds, low levels of virus may persist, and alteration in management or in the viral genome may cause new epidemics to occur [43,45]. The practise of preventing outbreaks using feedback may be a way to maintain not only the virus but also other pathogens in the herd [43,45,92].

NPD is of increased importance in intense production systems [34], but overt disease results from a perturbed balance between the individual’s resistance and the pathogenic potential of the microbe [57]. In the neonate, the alimentary tract is flooded with bacteria, probably due to the high pH of the stomach [33], and the outcome depends on the innate and passive immunity, environmental conditions, and the infectious pressure [22,27]. Further, nursing pigs have a functional defect in NK cells and an immature immune system [90]. A poor passive immunity may occur in herds with a high turn-over rate of sows and many gilts, or in sows with agalactia, resulting in inept composition of the antibodies provided [34]. However, colostrum alone is not always sufficient to protect the young piglets [50]. Negative environmental conditions (e.g., draught, chill, or poor management) or a high pathogen load (caused by, e.g., poor hygiene, continuous production systems, dual infections, and high humidity) may thus overcome the individual’s capability to sustain a challenge [22,29,34,50,99].

## 11. Conclusions

Most investigations target a list of predefined pathogens, thereby excluding the concept of an unprejudiced study design. In the research as well as in clinical practise, the basis to elucidate the aetiology behind NPD and achieve a correct diagnosis must be the necropsy of adequate cases. Thereby, relevant lesions, and subsequently, the pathogen causing these lesions, may be identified, enabling appropriate treatment. In the case of previously unrecognised presumptive pathogens, the fulfilment of Koch’s postulate is still of significant value.

Based on the current knowledge, ETEC still seems to be a valid cause of diarrhoea in single, 1to 5-day-old piglets, although its overall importance has decreased. TGE seem to be outdated, whereas single outbreaks of PED may occur. Currently, there is no evidence that CpA is a primary cause of NPD. Further, the diagnosis of *C. difficile* should be interpreted with caution and must include the presence of colitis and typical “volcano lesions”. Rotavirus causes diarrhoea in piglets ≥5 days of age, but due to its common presence in healthy animals, conclusive diagnosis should be based on the demonstration of the virus in the presence of typical lesions. *C. suis* is a common cause of diarrhoea in pigs >1 week of age, but optimal conditions for sporulation may also cause disease in younger pigs. Further, accumulating evidence suggests the involvement of certain strains of *Enterococci* spp. in NPD. It should, however, be acknowledged that the literature forming the basis for these conclusions mainly originates from Western pig-producing countries, and the situation in other countries may be different, although comprehensive surveys are currently lacking. It is also possible that there are emerging pathogens that may cause neonatal diarrhoea without being detected by the diagnostic methods employed.

## Data Availability

All data included in this text is given in the list of references. The references may be found on the internet, or may be provided by library services. Upon request, single papers may also be provided by the author.

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
