# Peer review of "On the Infectious Causes of Neonatal Piglet Diarrhoea—A Review"

_vetsci, 2022, doi:10.3390/vetsci9080422_

Round 1
Reviewer 1 Report
Thank you for having me review this fine overview of potential pathogens involved in porcine neonatal diarrhoea, which is a very relevant topic for health, welfare, antimicrobial consumption and prevention of disease in modern pig production. I enjoyed reading the paper, but I also have some suggestions for improvement:
General comments:
Title:
The title states “aetiology of neonatal piglet diarrhoea” which makes me wonder whether neonatal diarrhoea always has an infectious background, or could other factors be of importance? i.e protein level of the sow feed? Otherwise perhaps include ‘infectious’ as part of the title.
Abstract: L 15 perhaps add ‘field based’ or ‘microbiological’ to surveys (see later coment).
Part 1. Background (and introduction): A thorough historical introduction to the topic is given.
I lack a part explaining the clear aim of this paper. Both regarding topics and papers included and excluded in the review, but also what defines the surveys selected and presented. A short explanation is present in the abstract, but explained in more detail.
What is the methodology used? (structural literature search, databases, key words?).
Part 2. Clinical signs: This is a strict list of clinical signs of the different agents described in this paper: Perhaps some of the clinical described for Enterococcus spp in part 5 and Rotavirus in part 6, should be moved to this part? Alternatively, this section could be rewritten focusing on the clinical signs and pathogenesis instead of agents i.e. the mechanisms behind diarrhoea.
Part 3. E.coli: I read about intimin at some point: Is that of importance?
Bacterial agents in general: Was Salmonella ever described in neonatal piglets?
A general suggestion for the part about infectious causes: The section contains many abbreviations: Would It be possible to do at table with these explained? At the same time sorting and explaining disease groups, toxins and adhesion characteristics?
Table 1: This is probably a layout-issue: At first, I did not recognize that E.hirae belonged to the Swedish study (which I could see in the text).
L 6-12 + L 490 - onwards are not filled in: I assume that is part of the editorial process.
Technically, the paper is well written, and I have minor comments only, regarding spelling:
L64 became -> become
L140 + L174 caecum?
Author Response
Comments to Reviewer #1
Thank You for the valuable comments. We sincerely appreciate the efforts and the comments have been very helpful in improving our manuscript.
The importance of the sow feed for the milk composition and the subsequent development of diarrhoea has been debated from time to time, e.g. in the 1980s Tom Alexander put forward the theory that a high content of fat in the milk would partly constitute the aetiology behind “milk scour”. To our knowledge, there has been no research since then that unambiguously have proven that the milk composition per se would cause piglet diarrhoea. However, to avoid any mistakes to be made by us, the word “infectious” has been included in the title as suggested.
The word “microbiological” has been included in the abstract as suggested.
The aim has been included and extended, and the methodology used has been included under a separate heading; 1. Methodology. The subsequent headings have been re-numbered consecutively.
Thank you for this valuable suggestion. The clinics described in formerly section “5. Enterococcus spp.” and section “6. Viral infections” has instead been included in current section “3. Clinical signs”.
Formerly part “3- E. coli”. Yes, you are completely right, intimin is an important adhesion factor, causing the intimate attachment to enterocytes, and this information has been included in the description of the EPEC strains (Line 178).
Salmonella spp. are generally not considered as a cause of neonatal porcine diarrhoea, although there is one report on neonatal pigs developing diarrhoea following the experimental inoculation with S. Typhimurium respective S. Cholerasuis. Otherwise, pigs reported to suffer from diarrhoea caused by Salmonella spp. are either older (commonly, weaned pigs), or the age of the pigs is not clearly stated (reference no. 41 in the current manuscript). Therefore, we decided not to include salmonellosis among the agents described.
Thank you for this suggestion. A list of abbreviations has been included in the beginning of the manuscript. However, although we realize the need for it, we hesitate to include more information on the sorting of the different disease groups, toxins and adhesion factors for several reasons: Firstly, there is no definite consensus among the researchers on the definition of the various groups, nor on the genes defining the pathogenicity. Secondly, except for ETEC and oedema disease, most of these definitions are based on human studies, studies made on pigs are few and the pathotypes are mostly extrapolated from the human studies, thus, we cannot be completely sure that they holds true also for pigs. For these reasons, we have commonly used wordings such as “may”, information that would be lost in a table. Our intention with this review is to give an overview of the NPD complex as a whole and therefore, the information on each specific, disease-causing agent is brief. Thus, to avoid an overload due to the extended information that ought to be provided in such a table, readers that are specifically interested to dig deeper into the taxonomy of e.g. E. coli may instead read the references provided.
Table 1: For readability, “Table 1” was oriented in the “landscape mode” and was therefore not placed in the main text as suggested in the “instructions to the author”, but instead embedded as a zip file in the manuscript. We cannot explain the differences encountered but we have tried to increase the readability of the table in the manuscript. However, we agree that this should be an editorial issue.
Formerly line 64: The word “became” has been changed to “become”.
Formerly line 140 and line 174: The spelling of “cecum” has been changed to “caecum”.
Line 6-12 and line 490 onwards: This information has been included in the edited version.
Reviewer 2 Report
The neonatal diarrhea is a big issue on swine production and its etiologies are very diversified. This is certainly an ambitious plan to review, however, no one can cover this issue completely unless focused on a specific aspect of the neonatal diarrhea.
line 58: the major pathophysiological mechanisms of diarrhea include hypersecretion, malabsorption, and inflammation. Here authors did not mention "inflammation". Is the "inflammation" usually does not account for the neonatal diarrhea?
line 91: spell out "Coccidia". The C. here may be confused with Clostridium.
sections 1-11: these information can be found in the literature. Authors have done a good summary on the historical development of neonatal diarrhea. However I suggest authors cite more references published after year 2018, and focus on the "clinical aspect" providing an algorithm for a swine medicine clinician on how to approach from the clinical signs of "neonatal diarrhea" to the diagnosis of etiology and treatment.
Sections 12 and 13 are the merits this manuscript. To me, this seems to be the impression on the evolving situation according to her observation over the past years in Sweden. This may be viewed as a "regional and personal" experience, (which is good to know) but the evolution of situation (line 414) may not be the same elsewhere because of different swine raising environment. If this is true, author should emphasize this in the conclusion.
Table 1 and line 473-475: Table 1 is ambitious. But as the author emphasizes, the basis for studies of the etiology behind NPD must be the necropsy, lesions, and pathogen identification. So what published in the literature is limited by the capacity of the laboratory diagnosis, and this may not reflect the real situation in the field (what the clinical observed).
line 453: I disagree with that" coronavirus is a seeming of less importance today". Based on the recent manuscripts that I reviewed, coronaviruses-induced neonatal diarrhea is a hot issue, at least in USA and PRC (but again this is based solely on the literature) perhaps because of different raising environments including intensity. Research on PEDV, and swine acute diarrhea syndrome coronavirus (SADS-CoV) as differential diagnoses for TGEV are hot.
Author Response
Comments to Reviewer #2
Thank You for the valuable comments. We sincerely appreciate the efforts and the comments have been very helpful in improving our manuscript.
Formerly line 58: Yes, we certainly do agree that inflammation is part of the pathophysiological mechanisms behind diarrhoea. However, the definitions given in this section are the definitions found in the literature on the clinical sign “diarrhoea”. We agree that some of the definitions also touch upon the pathophysiology, but in our review we may only cite the definitions as they are given, although one may speculate that these definitions probably reflect the difficulties in finding objective ways to measure and describe a clinical sign. We have tried to clarify that the definitions refer to the sign “diarrhoea” only, by highlighting it by quotation marks.
Formerly line 91: Thank you for this comment. The coccidian species Cystoisospora suis has been spelled out as requested (line 143). To increase the readability of the text, we have also included a list of abbreviations in the beginning of the manuscript.
Sections 1-11: We believe this to be a very good suggestion. However, we have found very few, recent scientific papers that focus on the clinical aspects of NPD. This is probably because the most thorough clinical descriptions often are presented at the first observations of a certain disease, which partly is the reason why we included the “background section”. Further, as several authors pointed out (references nos 34, 41, 43, and 74), the clinical signs are rarely indicative of a certain aetiology. The best way to approach the diagnosis of neonatal diarrhoea of uncertain aetiology is the necropsy of adequate specimens, followed by targeted sampling. We have further stretched this in the first section under the heading “11. Conclusion”.
Sections 12 and 13: Thank you very much for this very benevolent comment. We do not, however, believe that this merely reflect the evolving situation in Sweden. A similar trend has been noted in many, important swine-producing countries: Spain (ref. nos 27, 65, 118), Germany (ref. no 74), Denmark (ref. nos 16, 25, 85, 86, 119), Canada (ref. nos 3 and 43), the US (ref. no 113). However, we agree that this may not be the case in other pig-producing parts of the world, although few reports to substantiate this are available. We have thus added this information in the conclusions.
Table 1 and line 473-476: We do agree! This is the reason why we choose to point out these limitations. We have tried to include the aspects in the discussion, since they should be accounted for in any future studies. As already discussed, a diagnosis based solely on the clinical signs are not very reliable. However, it is possible to perform a necropsy on site, collecting samples for histology using very limited equipment, thereby substantially improving the diagnosis, regardless of the laboratory analytic methods available.
Formerly line 453: Thank you for this comment. We agree that PED may still be of a large importance in some parts of the world, since the last outbreaks reported occurred in 2015, and we also agree that SADS-CoV seem to be an emerging disease in China. We have further stretched this by including a review from 2018 on the presence of coronaviruses in China in the list of references (no 89). Accordingly, we have changed the sentence to merely refer to TGE.
Round 2
Reviewer 2 Report
lines 60-65, Table 1, line 541, reference 89: I think the method of you literature search limited the publications to mostly before the year 2018 (Table 1), despite the addition of reference 89 (2019). The search seemed to miss most literature published in between 2019-2022, reporting single emerging diseases, that are not necessarily indexed by those key words you used. It is really nothing to do with ""developing countries".
Author Response
Dear Sir.
lines 60-65, Table 1, line 541, ref. no 89: Thank you for your comment. However, using the search words provided and also performing an additional search for emerging diseases including the search words: "neonatal piglet diarrhoea" OR "Diarrhoea syndrome" AND "pigs" AND "coronavirus" OR "emerging diseases" AND "2019-2022", we have not been able to identify any papers providing additional information to meet the goal of our review. We apologize for this but it is unclear to us how we should proceed?
If you are referring to a paper describing the first mention of a new coronavirus isolated from cases of neonatal porcine diarrhoea in pigs, this information should be included in reference no, 87-90, or it should be covered by the sentence "However, new mutation and recombinant variants occur" . To our knowledge, the coronaviruses currently identified to cause neonatal diarrhoea in pigs includes TGE, PED, deltacoronavirus and SADS-CoV that are all included in ref. nos 87-90. Are there any additional coronaviruses or any other, single emerging diseases causing NPD, that we have not been able to identify in our review?
If so, would it be possible for you to specify the literature that you would like us to include in our paper?
We have deleted the text " e.g. developing countries" as suggested.